# How do Human Processes AI-generated Hallucination Contents: a Neuroimaging Study

## Abstract

Hallucinations produced by multi-modal large language models (MLLMs) pose considerable risks as it remains unclear to what extent humans can accurately recognize them. To address this issue, this paper explores humans' neural responses to such hallucinated content across varying time scales. We record EEG from 27 participants while they are viewing contents generated by a multi-modal large language models that either include hallucination words or not, and judge whether each description matched an image. The collected EEG data is analyzed based on averaged event related potentials (ERP) on hallucination vs non-hallucination words. Results suggest that multiple cognitive processes, e.g., semantic integration, inferential processing, memory retrieval, and cognitive load, are engaged during humans' recognition of hallucination content. However, when hallucinations are not recognized by human participants, the brain treats them no differently from non-hallucination content. This indicates that humans already treat such hallucinations the same as non-hallucination content at a subconscious level. Furthermore, we conduct a prediction experiment that uses the collected EEG to detect hallucination contents. This indicates that we can detect whether a user has been deceived by hallucinations generated by MLLMs with their brain activities.

## 1 Introduction

Over the past few years, multi-modal large language models (MLLMs) have made impressive progress, scaling in size, architecture sophistication, and capability (Team (2025)). These advances have enabled them to perform a wide spectrum of tasks, from image captioning and generation to open-ended conversation and multi-modal understanding. However, one of their key drawbacks, hallucinations, i.e., the tendency to generate content that is plausible in surface form but factually incorrect or ungrounded, has become a growing concern.

Recently, researchers have attempted to understand the origins of hallucinations in MLLMs and LLMs (Ji et al. (2023); Huang et al. (2025)). Much of the existing research has approached hallucinations from the model perspective: exploring how aspects such as the training data, prompt design, decoding or sampling strategies, and internal uncertainty or confidence measures contribute to the occurrence of hallucinations (Maynez et al. (2020)). To mitigate the harmful effects of hallucinations, existing studies have been working on detecting hallucinated content automatically and further deriving mechanisms to generate factually consistent content. For example, introduced retrieval-augmented generation has been introduced to augment the models with factual knowledge. proposes to use several verification steps during generation to check the consistency of generated content (Farquhar et al. (2024); Zhang et al. (2024); Lewis et al. (2020)).

As MLLMs become more powerful and ubiquitous, many users have come to trust their outputs quite heavily, even when those outputs may be incorrect and misleading. This over-reliance on model-generated content can have negative consequences (Sun et al., 2024). Studies in human-AI interaction have begun to highlight these risks, showing that trust in LLMs can lead to undesirable effects on decision making and user behavior (Klingbeil et al. (2024); Zhai et al. (2024); Kim et al. (2025)). However, these approaches rely on explicit, post-hoc judgments rather than ongoing, automatic signals of perception or recognition from human observers (Barros (2025)). Some detection methods also focus on internal model states rather than how humans process hallucinated content in real time.

However, few studies have examined, from a neuroscience perspective, how human brain activity patterns differ when viewing hallucinated versus non-hallucinated content generated by multi-modal large language models. Understanding the neurological mechanisms behind it can illuminate which processing stages (early perceptual/attention, semantic integration, or late decision) play critical roles in detecting hallucinated content. By identifying neural patterns that precede behavioral judgments, such research can allow prediction or early warning of hallucinations beyond what accuracy and reaction time alone can reveal.

In this paper, we aim to fill the gaps identified above, investigate the neurological mechanisms by which humans recognize hallucinations, and gain insights for the development of LLMs. Specifically, we have the following research questions.

- **RQ1.** Do patterns of brain signals show significant differences when participants are viewing hallucination words versus non-hallucination words?

- **RQ2.** If yes, is this difference modulated by whether the participant correctly recognizes a hallucination word?

- **RQ3.** Can we predict whether AI-generated content contains hallucination based on EEG signals collected during viewing it?

To address these research questions, we collected EEG data from 27 participants. Each participant viewed textual stimuli generated by an MLLM that included both hallucinated and non-hallucinated content. On the basis of this paradigm, we conducted averaged event-related potential (ERP) analyses. An ERP is like a "brain fingerprint" that presents the patterns of brain signals in different groups (i.e., hallucination vs non-hallucination). The ERP analysis reveals a significant difference in brain signals when participants are processing hallucination words and non-hallucination words.

We further decompose the detected ERP difference into different temporal scales. Results reveal that multiple cognitive processes, such as semantic-thematic integration, inferential processing, memory retrieval, and cognitive loading, are engaged in hallucination recognition. However, when participants failed to detect AI-generated hallucination, we did not observe characteristic neural signatures of anomaly detection. This neural "silence" reflects endogenous cognitive mechanisms that attenuate conflict detection when the hallucinated content appears fluent and contextually plausible. We speculate that such fluency reduces the engagement of discrepancy-monitoring processes, thereby making the hallucinated information easier to accept without triggering strong neural alarm signals. Additionally, we show that EEG signals can be used to predict, at both the word-level and sentence-level, whether content contains hallucination. This prediction is reliable only when participants correctly recognize hallucination. In other words, AI-generated hallucinated content has the potential to deceive users from neural processing all the way to behavioral outcomes.

## 2 DATA COLLECTION

In this section, we describe the collection of EEG and behavioral data from our 27 participants while they completed the multimodal QA task designed to probe hallucination recognition.

### 2.1 PARTICIPANTS

A total of 27 volunteers were recruited for this study, comprising 11 males and 16 females, aged between 19 and 30 years (with an average of 24). The sample comprised mostly college students—but also included several members of the general public, ensuring some diversity beyond the academic population. The participants represented a range of disciplines (e.g., computer science, mechanical engineering, chemistry, and environmental engineering), spanning undergraduate to postgraduate levels. Each individual completed the full experiment in approximately 1.5 hours, which includes 30 minutes for equipment setup and task instructions. Prior to participation, all individuals were informed that their time would be compensated at a rate equivalent to US$11.8 per hour, contingent upon their completion of the study, to ensure the quality of the data collected for the study.

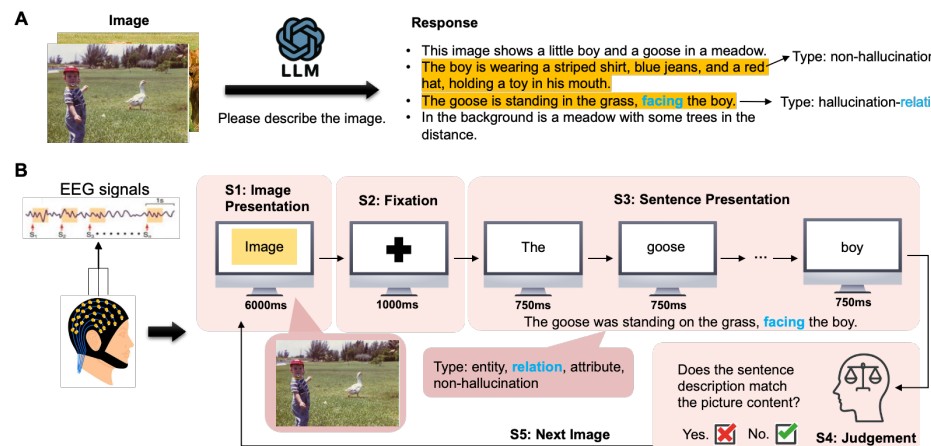

Figure 1: The overall procedure of our data collection. A) The procedure of stimulus selection. B) The experimental trial flow consists of five stages: presenting an image (S1), showing a fixation cross (S2), displaying a sentence word-by-word (S3), the participant making a judgment about the sentence's match to the image (S4), and finally proceeding to the next image (S5).

## 2.2 TASK PREPARATION

To minimize bias arising from participants' varying disciplinary backgrounds, we deliberately adopted a multimodal QA task that demands minimal prerequisite knowledge. Our approach draws on the AMBER benchmark (Wang et al. (2023))—a multi-dimensional, LLM-free evaluation dataset for hallucination in MLLMs—which comprises 1,004 images derived from the MSCOCO (Lin et al. (2014)) and includes detailed annotations for hallucination at the levels of existence, attribute, and relation. Using this benchmark, we generated responses to generative-style prompts for each image via a MLLM. The MLLM we chose in our study is Qwen2.5-VL-3B-Instruct (Team (2025); Wang et al. (2024)). Leveraging both the original hallucination annotations provided by AMBER and our own manual verification, we selected 60 image–response pairs where the MLLM clearly exhibited hallucinatory content. For each of these responses, we extracted one sentence containing a hallucination and one sentence free of hallucination, thus forming balanced stimuli for EEG testing. To ensure that each sentence we selected is not an illusion in itself, we use GPT4-HDM method Su et al. (2024) and input each sentence into an LLM separately to let it judge whether it violates the common sense of the real world. Following AMBER's taxonomy of hallucination types, we categorized the stimuli as follows: 27 entity-related, 13 relation-related, and 26 attribute-related (attribute category further subdivided into action (5), count (11), and state (10)). Representative cases and the full selection criteria are documented in our publicly accessible code repository. The procedure of stimuli selection is shown in Fig 1 A.

## 2.3 PROCEDURE

Before the main trials, participants first completed an entry questionnaire and signed informed consent regarding privacy and data security. They then received detailed instructions explaining the primary tasks and the operational procedures, and were explicitly informed that they retained the right to withdraw from the study at any time without consequence. Following orientation, participants carried out a series of training trials intended to help them become familiar with the formal experiment's flow. Each participant was also asked to select a random seed before the experiment began, which was used to randomize the order of stimulus categories in order to ensure that across participants, each hallucination type and image condition would be fairly and evenly presented.

Once these preparatory steps were finished, each trial proceeded through stages S1 through S5 in sequence, as shown in Fig 1 B. In S1 (Image Presentation), an image is shown for 6000 milliseconds while participants are told to view it attentively, knowing there will be a later match judgment. In S2 (Fixation), a central fixation cross is displayed for 1000 milliseconds to orient and stabilize visual attention. In S3 (Sentence Presentation), a sentence description appears word by word: the first

word (e.g., "The") is shown for 750 milliseconds (Ye et al. (2022)), followed by each subsequent word for the same duration; the sentence may either contain a hallucination of a different type or be non-hallucinated. After the full description, in S4 (Judgement) participants are asked whether the sentence matches the image content via a binary choice (Yes or No), responding using key presses. Finally, S5 introduces the next trial and cycles back to S1 when participants press the space key. During the entire experiment, we continuously recorded EEG signals from each participant. Using event triggers, we logged the onset times of all key stimulus events, so each segment of EEG could be aligned precisely to the relevant experimental stage. In addition, for every single sentence shown, we recorded the participant's judgment (Yes/No) about whether the description matched the image.

## 3 Result Analysis

In this section, we employ ERP analysis techniques to investigate how brain signal patterns differ when participants view hallucination versus non-hallucination words, and synthesize these findings to outline the neural mechanisms by which humans recognize hallucinations. The detailed code can be accessed openly through the url `https://anonymous.4open.science/r/Neural-Correlates-of-AI-generated-Multimodal-Hallucinations-2F7D`.

### 3.1 Statistic Analysis

Across the full set of 120 trials per participant, on average participants answered 101.14±6.53 items correctly, yielding a mean overall accuracy of 84.29%. Breaking this down by condition, for the 60 non-hallucination trials the average accuracy was 81.17%, while for the 60 hallucination trials it was higher, at 87.41% — participants thus performed better on sentences containing hallucination words than on non-hallucination ones. Considering hallucination categories, the mean recognition accuracy by type was relation: 90.88%, entity: 89.30%, and attribute: 86.18%. Statistical tests reveal that the accuracy across all hallucination categories do not differ significantly. While the accuracies across these categories are fairly similar, the relation type had the highest performance and the attribute type the lowest, suggesting that relation-based hallucinations may be easier for participants to detect, whereas attribute-level hallucinations pose greater detection difficulty.

### 3.2 ERP Analysis

ERP refers to brain voltages that are time-locked to specific events and reflect neural responses elicited by those events (Blackwood & Muir (1990)). One of its key advantages is the high temporal resolution it offers, and the sequence of ERP peaks provides precise insight into rapid neural processing stages (Luck et al. (2000)). ERP components are evoked amplitude in different post-stimulus time windows, e.g., N100, N400 (negative waves within 100ms, 400 ms), and P200, and P600 (positive waves within 200 ms, 600 ms). These standard ERP markers index different cognitive operations (Luck et al. (2000)). In our analysis, we employ conventional ERP-processing procedures including signal preprocessing, defining time windows of interest, and specifying regions of interest (ROIs) for comparing conditions (Ye et al. (2022); Zhu et al. (2024); Ye et al. (2024)). The method we use to preprocess the EEG signals is detailed in Appendix A.2.

To disentangle different ERP components, we partitioned the extracted time interval into several distinct time windows based on the approach introduced by Lehmann & Skrandies (1980). Their method identifies evoked scalp potential components by examining both their latency and their topographic pattern. In our analysis, we computed the Global Field Power (GFP) over the 50-750 ms post-stimulus interval, and delineated time windows around the GFP peaks, as shown in Table 1.

To facilitate subsequent analyses, we partitioned the EEG data according to both the stimulus word type and participants' recognition performance. We defined three categories: **Hallu** for hallucination words that the participant correctly recognized as hallucinated; **NoHallu** for non-hallucination words correctly identified as non-hallucinated; and **HalluWrong** for hallucination words which participants failed to recognize (i.e., words that were in fact hallucinations but were judged as non-hallucinations). We plot the grand-average ERP waveforms for different stimulus word types in central brain region in Figure 2.

Table 1: The statistical significance test results for different ERP components across brain regions for Hallu vs. NoHallu words. We use the repeated measures ANOVA test and adopt post-hoc pair-wise comparisons with FDR correction. * and ** indicate statistical significance at a level of p<0.05, p<0.001, respectively.

| Time window | ROI | RM-ANOVA test | Post-hoc test |
|---|---|---|---|
| 50–120 ms | r-temporal, parietal | Hallu >NoHallu * | * |
| 120–280 ms | pre-frontal, occipital | Hallu >NoHallu * | * |
| | frontal, central, l-temporal | Hallu >NoHallu ** | * |
| 280–550 ms | r-temporal | Hallu >NoHallu * | * |
| | central | Hallu >NoHallu ** | * |
| 550–750 ms | pre-frontal, frontal, l-temporal, r-temporal, occipital | Hallu >NoHallu * | * |
| | central | Hallu >NoHallu ** | ** |

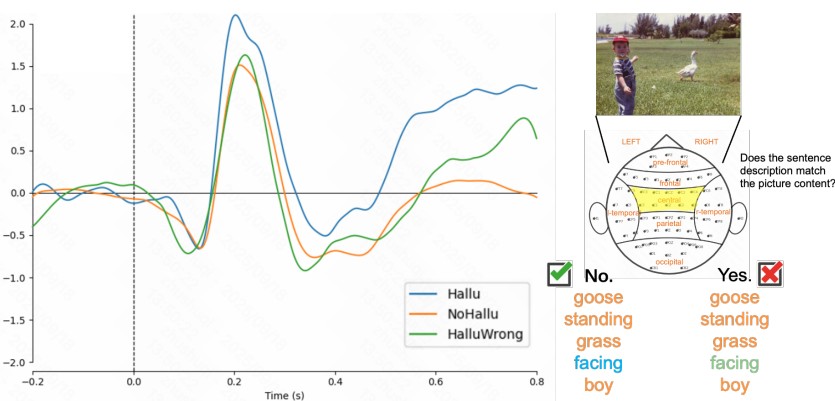

Figure 2: Comparison of ERP waveforms for different stimulus word types in the central brain region.

### 3.2.1 HALLU VS. NOHALLU WORDS

We divide the electrodes into seven brain regions according to their placement on the brain topography shown in Figure 2. We applied a one-way repeated-measures ANOVA in a fixed time window for each brain region, followed by post-hoc pair-wise comparisons with FDR correction. The statistical findings for the various time windows and regions of interest for Hallu vs. NoHallu words are presented in Table 1. Below, we discuss the characteristic features of each component and their potential functional roles.

N100. N100 is an early component in time window around 100 ms (50–120 ms). We employ repeated measures ANOVA method and discover significant differences between the grand-averaged N100 component in r-temporal (F[1,26]=4.615, p<0.05, $\eta_p^2$=0.151) and parietal (F[1,26]=4.872, p<0.05, $\eta_p^2$=0.158). The N100 component is typically interpreted as reflecting very early visual perceptual processing, especially for low-level visual features (Yang et al. (2022)). It often shows maximal expression in occipito-parietal regions. Recent work also indicates that the amplitude of N100 is closely linked with attentional allocation. Larger N100 amplitudes have been observed when stimuli draw more attention, or when perceptual systems are required to allocate greater resources to processing salient or unexpected input (Thornton et al. (2007); Rutman et al. (2010)). The results of statistical significance testing indicate that during the recognition of hallucination words, participants show enhanced N100 responses compared to non-hallucination words. This suggests that the process of identifying hallucination words recruits attention very early and imposes a higher cognitive load on the perceptual system, even before later semantic processing stages.

**P200.** P200 is the dominant component in time window around 200 ms (120–280 ms). RM-ANOVA reveals the significant differences between grand-averaged P300 component in pre-frontal (F[1,26]=8.575, p<0.05, $\eta_p^2$=0.248), occipital (F[1,26]=11.246, p<0.05, $\eta_p^2$=0.302), and frontal (F[1,26]=18.226, p<0.001, $\eta_p^2$=0.412), central (F[1,26]=15.311, p<0.001, $\eta_p^2$=0.371), l-temporal (F[1,26]=14.037, p<0.001, $\eta_p^2$=0.351). The P200 component is generally understood to reflect early attentional engagement and decision-related processing. It has been associated with novelty detection, stimulus complexity, and perceptual salience, such that more complex or unexpected stimuli elicit larger P200 amplitudes (Ghani et al. (2020)). Empirical work shows that P200 amplitude tends to increase with attentional load and with stimuli that violate perceptual or contextual expectations (Kemp et al. (2009); Polich (2007)). We observe enhanced P200 responses for hallucination words compared to non-hallucination words. This suggests that hallucination words impose greater demands on early stimulus selection. The processing system flags such words as perceptually or lexically salient, because they diverge from semantic expectation or evoke conflict.

**N400.** N400 component is evoked around 400 ms after the stimulus (280-550 ms). Significant differences are found in central (F[1,26]=16.442, p<0.001, $\eta_p^2$=0.387), r-temporal (F[1,26]=7.929, p<0.05, $\eta_p^2$=0.234). The N400 component is widely understood to index the access and integration of semantic information. It typically reaches its maximal amplitude at centro-parietal electrode sites, reflecting the brain's effort to reconcile a word's meaning with its broader context. Empirical findings show that less predictable or semantically incongruent words evoke larger N400 responses, consistent with the idea that the N400 is sensitive to violations of expectation and relates to retrieval from semantic memory (Lau et al. (2008); Lindborg et al. (2023); Michaelov et al. (2022)). Hallu words conflict with the visual context, hence they elicit greater N400 amplitudes than NoHallu words. This suggests when the descriptive content generated by the model diverges from what is visually present or semantically expected, the cost of semantic integration increases.

**P600.** P600 waveform mainly appears in time window around 600 ms (550-750 ms). Through ANOVA, we find significant differences between grand-averaged P600 component in pre-frontal (F[1,26]=5.874, p<0.05, $\eta_p^2$=0.184), frontal (F[1,26]=7.268, p<0.05, $\eta_p^2$=0.218), l-temporal (F[1,26]=7.517, p<0.05, $\eta_p^2$=0.224), r-temporal (F[1,26]=8.093, p<0.05, $\eta_p^2$=0.237), occipital (F[1,26]=6.561, p<0.05, $\eta_p^2$=0.202), and central (F[1,26]=31.558, p<0.001, $\eta_p^2$=0.548). The P600 (or late positivity) component is classically implicated in sentence processing tasks and shows its strongest responses at centro-parietal electrode sites. Originally, the P600 was discovered as an index of syntactic reanalysis and repair, reflecting efforts to restructure or repair comprehension (Seyednozadi et al. (2021)). More recently, research has shown that even in sentences that are grammatically correct, semantic conflict or non-typicality can also provoke a P600, which is referred to as the "semantic P600" (Bornkessel-Schlesewsky & Schlesewsky (2008); Brouwer et al. (2012)). In the context of hallucination recognition, once a word is identified as semantically hallucinatory, participants engage controlled reanalysis and decision/monitoring processes. These processes recruit language-time systems and posterior integration networks, consistent with the late positivity seen in P600. Thus, detection of hallucination involves not only early sensory/attentional and semantic mismatch stages, but also later re-evaluation and integration when the linguistic input conflicts with perceptual or expectation-based models.

### 3.2.2 HALLUWRONG VS. NOHALLU WORDS

We applied the same statistical analyses—comparing across multiple time-windows and regions of interest—to ERP responses elicited by HalluWrong versus NoHallu words. Across all examined components and brain regions, none of these comparisons reached statistical significance. Full test statistics for each time window and ROI are provided in the Appendix A.3. Importantly, we conducted a post-hoc sensitivity analysis to evaluate the detectability of HalluWrong-specific effects under our design since the number of HalluWrong trials was smaller. The within-subject effect size for the HalluWrong versus NoHallu contrast was small ($d_z$=0.126), and the corresponding post-hoc power to detect such a small effect at $\alpha$=0.05 was only 0.31. In contrast, the same analysis indicates that our sample size provides approximately 80% power to detect medium-to-large effects ($d_z \geq$0.372). This indicates that there exists no medium-to-large ERP effects when comparing HalluWrong versus NoHallu words.

## 3.3 DISCUSSION

Overall, our findings advance understanding of the neural mechanisms by which humans recognize hallucinated content generated by MLLMs. The ERP results clearly show that various cognitive processes are engaged at extremely fine temporal scales. Specifically, differences in early perceptual attention and cognitive load (P200/N100), semantic-thematic understanding (N400), inferential processing, and memory retrieval (P600) mechanisms underlie successful hallucination recognition (addressing **RQ1**). These observations are consistent with prior studies of comprehension mechanisms, which posit that unexpected or incongruent input requires more effortful retrieval and integration of meaning in memory and draws upon prediction error and expectancy effects (Zhu et al., 2024).

It is worth noting that our results align with, and extend, findings from previous ERP studies. For example,Ye et al. (2022) explored the neural mechanisms underlying reading comprehension, and Pinkosova et al. (2022) investigated relevance judgments. They both reported that answer words (or words highly relevant to the task) elicit larger ERP amplitudes compared to ordinary or low-relevance words. These patterns suggest that when a stimulus is more directly tied to achieving the experimental goal, participants tend to allocate more attentional and cognitive resources to those items. By analogy, our results suggest that participants in our user study tended to devote more resources to items that were more directly aligned with achieving the experimental task goal, i.e., detection of hallucinated content. While the precise mechanisms driving this attentional allocation remain beyond the scope of this paper, they represent an intriguing avenue for future research.

On the other hand, when participants failed to detect the hallucinated content, we did not observe the typical neural activity associated with anomaly detection. This neural "silence" may suggests that, due to their high linguistic fluency and contextual coherence, hallucinated outputs produced by advanced models can successfully evade the brain's automatic alerting mechanisms, thereby paving the way for the formation of false beliefs. This pattern implies that recognition (or conscious awareness) is a key trigger for abnormal neural responses. Mere exposure to hallucinated content does not suffice to induce the enhanced ERP effects (addressing **RQ2**). This aspect distinguishes the present task from many prior word-recognition or anomaly detection tasks.

The findings from our study offer valuable insights for AI and cognitive neuroscience. First, from the AI perspective, the discovery that neural patterns of hallucination only emerge when humans correctly recognize them suggests that truthfulness and consistency in models could be improved by incorporating mechanisms analogous to human recognition. These insights could guide the design of more robust hallucination detection and mitigation systems, especially in applications where human users rely on the model's outputs under limited oversight. Second, our study also suggests that the brain's ability to distinguish hallucinated content depends heavily on context and recognition, implying that studying hallucination purely from the model side may miss important human factors. For tasks involving domain-specific or expert knowledge, detection systems based on EEG or other neural signals will likely need to be collected among populations that have relevant background knowledge. Lastly, from a human-computer interaction standpoint, conscious awareness is essential for triggering anomalous neural responses, suggesting that user interfaces with active interventions may help reduce the risk of users accepting hallucinated content uncritically. Design strategies should emphasize helping users notice and identify when content seems incongruent.

## 4 PREDICTION EXPERIMENTS

To explore whether EEG signals can act as signals to predict whether the content generated by an MLLM contains hallucinations, we conducted word-level and sentence-level prediction experiments on the dataset we collected. In this section, we detail the procedures and results of experiments.

### 4.1 EXPERIMENTAL SETUP

**Task Definition** We formalize the prediction task as follows. Let a stimulus sentence contain $l$ words, and for each word, we extract EEG features during its presentation. We denote the sequence of word-level EEG features $X = [x_1, x_2, \ldots, x_l] \in \mathbb{R}^{l \times d}$ as input, where $d$ is the dimension of the extracted features. The model produces two outputs: word-level prediction vector $y_w = [y_{w,1}, y_{w,2}, \ldots, y_{w,l}] \in \{0, 1\}^l$ and sentence-level prediction $y_s \in \{0, 1\}$. For evaluation, we selected AUC as a metric.

Table 2: The classification results of word-level and sentence-level prediction. Best results are in **Bold**. †/* indicates the result is significantly different with p-value<0.05 compared to the best model and random, respectively.

| Models | word-level | | sentence-level | |
|---|---|---|---|---|
| | $AUC_{within}$ | $AUC_{cross}$ | $AUC_{within}$ | $AUC_{cross}$ |
| SVM | **0.9393**\* | **0.8631**\* | 0.9601†\* | 0.9494†\* |
| RF | 0.9069†\* | 0.7924†\* | 0.9622†\* | 0.9384†\* |
| GBDT | 0.9190†\* | 0.8362†\* | 0.9647\* | 0.8531†\* |
| MLP | 0.9125†\* | 0.8272†\* | 0.9655\* | 0.9824\* |
| attention | 0.9113†\* | 0.7955†\* | **0.9673**\* | **0.9846**\* |
| SVM (w. HalluWrong) | 0.5187† | 0.5033† | 0.5518† | 0.5391† |

**Feature Selection**   To build input features for our prediction models, we combined Frequency-Band-based Features (FBFs) with Event-Related Potential-based Features (ERPFs). FBFs capture global spectral information, while ERPFs focus on specific, behaviorally relevant time windows indicated by our ERP analyses. We selected four brain regions (central, l-temporal, r-temporal, and occipital) that consistently showed strong effects in our significance tests. For each of those four regions, we computed differential entropy for five standard EEG frequency bands. Differential entropy is widely used to quantify complexity in EEG signals, and has been shown to be effective for classification tasks such as emotion recognition (Chen et al. (2019); Duan et al. (2013)). Based on the ERP components that were significant in the previous section, we selected a set of time points within those time window and divided each into five equal segments. We concatenated the frequency-domain and the time-domain features to create a 760-dimensional input vector.

**Data splitting strategies**   To examine whether the model's performance is consistent across different participants and whether it generalizes robustly across varying participant data distributions, we employed two data splitting strategies. Within-subject paradigm means that for each participant individually, we perform ten-fold cross-validation on their data, and then average performance across folds. Across-subject paradigm means that we hold out one participant's data as the test set and train the model on the other 26 participants' data.

**Model selection**   We selected support vector machine (SVM), random forest (RF), multi-layer perceptron (MLP), gradient boosting decision tree (GBDT), and an attention-based model. Our rationale for not selecting more complex or highly specialized neural architectures is twofold: 1) this task is novel, and to our knowledge, no dedicated model has previously been designed specifically; 2) our goal in this work is to demonstrate the effectiveness of EEG as an implicit feedback signal for predicting hallucinated content. More sophisticated architectures to push maximal performance remain a promising direction for future work. For more model and training details, please refer to the code and Appendix A.5.

## 4.2 RESULTS

Table 2 presents the results of the word-level and sentence-level prediction classification. It shows that several models achieved strong performance, with SVM attaining the highest word-level performance and the attention-based model performing best at the sentence-level prediction. As expected, the cross-subject AUC scores are generally lower than the within-subject ones, likely due to inter-subject variability in EEG signals. Differences in brain anatomy, electrode placement, cognitive strategies, and noise make generalization across individuals more challenging (Apicella et al., 2024). Another consistent trend is that sentence-level classification outperforms word-level classification. This is plausible because sentence-level prediction allows the model to integrate information across all constituent words, capturing contextual dependencies and cumulative signals. The attention-based model in particular can exploit sequential dependencies via its internal weighting mechanism, which helps it better aggregate subtle signals across words. The results indicate that the EEG signals we collected carry meaningful information for predicting hallucinated vs non-hallucinated content. We further experimented by including HalluWrong words into the training data for SVM. The results

(shown in the last rows of Table 2) demonstrate a significant drop in performance, which indicates that only when participants correctly recognize hallucinations do the EEG signals carry discriminative information. Including misrecognized hallucination words will degrade the model's ability to generalize.

Overall, our experiments validate that, at both the word and sentence levels, EEG is a viable implicit feedback signal for detecting hallucination in generated content. However, this prediction is reliable only when participants correctly recognize hallucination. (addressing **RQ3**)

## 5 RELATED WORK

### 5.1 HALLUCINATION IN LLMS

Hallucination in LLMs denotes fluent but ungrounded or factually incorrect outputs (Ji et al., 2023). Prior work separates intrinsic drivers from extrinsic causes and advances two main strands: detection and mitigation. For detection, post-hoc verification with retrieval/KBs checks factuality in knowledge-intensive tasks (Lewis et al., 2020), while black-box consistency methods flag unstable generations without instrumenting the model (Manakul et al., 2023). AMBER offers an LLM-free, type-controlled benchmark, and object-level studies document captioning-specific hallucinations (Wang et al., 2023; Rohrbach et al., 2018). Mitigation commonly uses retrieval-augmented generation (RAG) to inject verifiable evidence during decoding (Lewis et al., 2020). Complementary strategies include self-critique and verifier pipelines to reject dubious claims and improved cross-modal alignment in MLLMs to curb object and attribute hallucinations (Manakul et al., 2023; Wang et al., 2023; Rohrbach et al., 2018). Despite progress, most evaluations remain outcome-based, leaving open when and how humans neurally register hallucinations.

### 5.2 NEUROSCIENCE & AI

Event-related potentials (ERPs) provide time-resolved markers of cognitive processing (Luck, 2014). N100 and P200 index perceptual and attentional allocation; P300 relates to task-relevant salience; N400 tracks semantic integration and expectancy violations; and P600 reflects late reanalysis and monitoring, including "semantic P600" effects (Bornkessel-Schlesewsky & Schlesewsky, 2008). Computational–neuroscience links show partial convergence between language-model representations and brain activity, and surprisal robustly correlates with N400 amplitude in sentence comprehension (Schrimpf et al., 2021; Frank et al., 2013). Beyond language, EEG studies of short-video polarization demonstrate that cognitive impact may be invisible to surface behaviors yet measurable in neural signals, and that EEG features can predict exposure to polarized content (Du et al., 2025a;b). Building on these insights, we compare ERPs elicited by hallucinated versus non-hallucinated words and condition effects on recognition.

## 6 CONCLUSION

In summary, this paper makes the following three contributions. 1) We collected and will release an EEG dataset from 27 participants, in which subjects viewed text generated by MLLM, including both hallucinated and non-hallucinated content. 2) We performed ERP analyses to probe the neural mechanisms of human recognition of MLLM-generated hallucinations and found that early attention and perceptual processing, semantic-thematic integration, inferential reasoning, and memory retrieval are all involved at very fine temporal resolution. Crucially, ERP differences between hallucination vs non-hallucination words only emerge when participants correctly recognize hallucination content, indicating that endogenous cognitive mechanisms that attenuate conflict detection when the hallucinated content appears fluent and contextually plausible 3) We demonstrated that it is possible to predict whether content contains hallucinations with EEG at both the word-level and sentence-level, but that reliable prediction depends on correct recognition of hallucinated content by participants.

Despite the promising findings, this study has several limitations that must be acknowledged. 1) Although we have a relatively large number of participants (n = 27), which helps statistical reliability, each participant in our dataset viewed relatively few hallucination words, since the EEG data collec-

tion equipment is not portable and the sessions are time-consuming. 2) Our setup was constrained to a laboratory setting. We made efforts to approximate real-world conditions, but there remains a gap between them. Factors such as ambient noise, participant movement, multitasking, and variations in attention in real life are not fully captured in our data.

Our experimental findings suggest that the prediction of hallucinated content may depend on whether participants have relevant domain-specific knowledge. In future studies, it would be valuable to conduct experiments within groups possessing specialized backgrounds (e.g. experts in medicine, law, science) to assess how prior knowledge modulates EEG signatures of hallucination recognition. This might reveal whether prediction models need to be tailored for knowledge domains, and whether neural markers differ across expert vs novice observers. Although our study examined several categories of hallucination (relation, entity, attribute), a more fine-grained investigation is needed to understand how different kinds of semantic and perceptual violations produce distinct neural effects. This would help map which categories are most difficult to detect, in which brain regions and at what latencies, thereby informing both cognitive theory and model design. Furthermore, while the dataset is sufficiently powered to detect medium-to-large effects, its sensitivity to very small neural differences may be limited. This constraint does not affect the robustness of our main findings, but it highlights the need for future work to employ longer multi-session recordings or adaptive paradigms that can more efficiently sample rare hallucination events.

## 7 ETHICS AND PRIVACY

To protect participants' privacy and physical health, our user study adheres to strict ethical guidelines for human research, with approval from the ethics committee[1]. In accordance with ethical standards, we have taken several steps to protect participants' privacy, including data anonymization and obtaining informed consent from all participants. Additionally, participants are thoroughly informed about the study's objectives, procedures, and potential outcomes. The EEG data collection method employed in this research is non-invasive and poses no risk to participants.

## 8 REPRODUCIBILITY STATEMENT

We are committed to full reproducibility of this work. The dataset will be publicly released after the review stage. All code, including data preprocessing, model training, and evaluation scripts, will be made available at github link. The architectural and implementation details of all models are documented in the Appendix A.5.

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

# A APPENDIX

## A.1 APPARATUS

The stimuli are presented on a desktop computer that has a 27-inch monitor with a resolution of $2560 \times 1440$ pixels and a refresh rate of 60 Hz. Participants are required to use the keyboard to interact with the platform. EEG signals are captured and amplified using a Scan NuAmps Express system (Compumedics Ltd., VIC, Australia) and a 64-channel Quik-Cap (Compumedical NeuroScan). A laptop computer functions as a server to record EEG signals and triggers using Curry8 software. Throughout the experiment, electrode-scalp impedance is maintained under $50k\Omega$, and the sampling rate is set at 1000 Hz.

## A.2 PREPROCESS

We preprocess the EEG data using several steps: first, we re-reference all recorded signals offline using the linked-mastoids method to reduce reference bias (Yao et al. (2019)); second, we apply notch, high-pass, and low-pass filters to eliminate environmental interference, slow voltage drift, and high-frequency noise respectively; third, we extract epochs of interest and compute their averages to obtain ERP waveforms. The epochs are defined from 200 ms before the presentation of each stimulus word to 800 ms after, covering the expected time window for relevant neural responses.

## A.3 ERP ANALYSIS

Table 3: Raw and FDR-corrected p-values (Hallu vs. NoHallu words)

|  | pre-frontal | frontal | central | l-temporal | r-temporal | parietal | occipital |
|---|---|---|---|---|---|---|---|
| **Raw p-values** | | | | | | | |
| 50–120 ms | 0.0609 | 0.2282 | 0.6916 | 0.1084 | 0.0082 | 0.0093 | 0.8691 |
| 120–280 ms | 0.0070 | 0.0002 | 0.0006 | 0.0009 | 0.0493 | 0.0731 | 0.0025 |
| 280–550 ms | 0.1148 | 0.0849 | 0.0004 | 0.0379 | 0.0092 | 0.5621 | 0.0304 |
| 550–750 ms | 0.0226 | 0.0121 | 0.0000 | 0.0109 | 0.0086 | 0.1085 | 0.0166 |
| **FDR-corrected p-values** | | | | | | | |
| 50–120 ms | 0.1422 | 0.3194 | 0.8069 | 0.1897 | 0.0326 | 0.0326 | 0.8691 |
| 120–280 ms | 0.0098 | 0.0016 | 0.0021 | 0.0021 | 0.0575 | 0.0731 | 0.0043 |
| 280–550 ms | 0.1339 | 0.1189 | 0.0028 | 0.0663 | 0.0321 | 0.5621 | 0.0663 |
| 550–750 ms | 0.0264 | 0.0213 | 0.0000 | 0.0213 | 0.0213 | 0.1085 | 0.0232 |

Tables 3 present the p-values before and after FDR correction, for different ERP components across brain regions, comparing HalluWrong vs. NoHallu words.

Table 4: Raw and FDR-corrected p-values (HalluWrong vs. NoHallu words)

|  | pre-frontal | frontal | central | l-temporal | r-temporal | parietal | occipital |
|---|---|---|---|---|---|---|---|
| **Raw p-values** | | | | | | | |
| 50-120 ms | 0.3095 | 0.1499 | 0.7489 | 0.7816 | 0.4652 | 0.2290 | 0.0801 |
| 120-280 ms | 0.2067 | 0.0775 | 0.4556 | 0.2885 | 0.8881 | 0.0796 | 0.0750 |
| 280-550 ms | 0.8963 | 0.8553 | 0.5562 | 0.5928 | 0.6642 | 0.4844 | 0.0958 |
| 550-750 ms | 0.8155 | 0.9407 | 0.2379 | 0.7053 | 0.6984 | 0.0924 | 0.3060 |
| **FDR-corrected p-values** | | | | | | | |
| 50–120 ms | 0.5416 | 0.5247 | 0.7816 | 0.7816 | 0.6513 | 0.5343 | 0.5247 |
| 120–280 ms | 0.3617 | 0.1857 | 0.5315 | 0.4039 | 0.8881 | 0.1857 | 0.1857 |
| 280–550 ms | 0.8963 | 0.8963 | 0.8963 | 0.8963 | 0.8963 | 0.8963 | 0.6706 |
| 550–750 ms | 0.9407 | 0.9407 | 0.7140 | 0.9407 | 0.9407 | 0.6468 | 0.7140 |

Tables 5 and Table 4 present the statistical significance test results (F scores and p values, respectively) for different ERP components across brain regions, comparing HalluWrong vs. NoHallu

Table 5: The statistical significance test results (F score) for different ERP components across brain regions for HalluWrong vs. NoHallu words.

| F[1,26] | pre-frontal | frontal | central | l-temporal | r-temporal | parietal | occipital |
|---|---|---|---|---|---|---|---|
| 50-120 ms | 1.0762 | 2.2069 | 0.1047 | 0.0786 | 0.5500 | 1.5207 | 3.3269 |
| 120-280 ms | 1.6805 | 3.3903 | 0.5743 | 1.1759 | 0.0202 | 3.3387 | 3.6863 |
| 280-550 ms | 0.0173 | 0.0340 | 0.3558 | 0.2935 | 0.1930 | 0.5038 | 2.9955 |
| 550-750 ms | 0.0556 | 0.0056 | 1.4622 | 0.1463 | 0.1536 | 3.0617 | 1.0923 |

**words.** The results indicate that for all ERP components and in all examined regions of interest, the differences between HalluWrong and NoHallu words are not statistically significant.

## A.4 MORE STATISTIC ANALYSIS

Table 6: All Participants' Number of Correct Items

| | | | | | |
|---|---|---|---|---|---|
| 100 | 85 | 106 | 106 | 98 | 100 |
| 93 | 97 | 112 | 103 | 104 | 101 |
| 90 | 103 | 106 | 96 | 107 | 98 |
| 88 | 111 | 104 | 105 | 102 | 107 |
| 106 | 101 | 102 | | | |

Table 6 presents the distribution of the number of correct items answered by all participants, where the mean is approximately 101.14, the standard deviation is 6.53, and the coefficient of variation is 6.46% — reflecting a relatively high level of consistency in the number of correct items among the participants.

## A.5 MODELS

The model structures and hyperparameters are as follows. For all models, the input features first undergo a preprocessing pipeline, which includes mean imputation for any missing values, followed by standard scaling to normalize the data.

SVM (Support Vector Machine) We use a Radial Basis Function (RBF) kernel. The regularization parameter $C$ is set to $1$. The model is configured to output probability estimates for classification.

RF (Random Forest) We set the number of trees in the forest to $100$. All other parameters are kept at their default values as specified in the scikit-learn library.

GBDT (Gradient Boosting Decision Trees) We set the number of boosting stages to $100$ and the learning rate to $0.1$. All other parameters are set to their default values.

MLP (Multi-Layer Perceptron) We implement a network using PyTorch. The architecture consists of a single hidden layer with $100$ units, which uses a ReLU activation function. A dropout layer with a probability of $0.5$ is applied after the activation function for regularization. The output layer is a linear layer that maps to the two output classes.

Attention-based model We use a Transformer Encoder architecture implemented in PyTorch. The input features are first projected into an embedding space with a dimension of $128$. This is followed by a 2-layer Transformer Encoder. Each encoder layer utilizes a multi-head attention mechanism with $8$ attention heads and a dropout rate of $0.5$. A final linear layer maps the encoder's output to the class scores.

Training Configuration for Deep Models For both deep learning models (MLP and Attention-based), we use the cross-entropy loss function and the Adam optimizer with a learning rate of $10^{-3}$. The models are trained for $300$ epochs with a batch size of $32$.

Table 7: The recall results of word-level and sentence-level prediction. † indicates the result is significantly different with p-value<0.05 compared to the best model. * indicates including HalluWrong words in the training.

| Models | word-level | | sentence-level | |
|--------|-----------------|-----------------|-----------------|-----------------|
|        | $Recall_{within}$ | $Recall_{cross}$ | $Recall_{within}$ | $Recall_{cross}$ |
| SVM | 0.5740† | 0.3669† | 0.6610† | 0.4655† |
| RF | 0.2629† | 0.1169† | 0.0404† | 0.0162† |
| GBDT | 0.4104† | 0.2132† | 0.4602† | 0.0346† |
| MLP | 0.6141† | 0.3487† | 0.8952† | **0.8244** |
| attention | **0.6617** | **0.4421** | **0.9310** | 0.7407† |
| SVM* | 0.1370† | 0.2340† | 0.2771† | 0.2349† |

### A.6    More Results

Table 7 shows the recall results of word-level and sentence-level prediction.

### A.7    The Use of Large Language Models

In this work, we leveraged large language models (LLMs) to assist in manuscript preparation, including refining the text for clarity and style, as well as facilitating literature retrieval. All LLM-generated suggestions were carefully reviewed, edited, and integrated by the authors to ensure scientific accuracy and consistency with our own writing voice. Meanwhile, the dataset on which our experiments depend was generated using an open-source multimodal large language model (MLLM), i.e., Qwen2.5-VL-3B. We adopted Qwen2.5-VL-3B to produce image-based descriptions from which we selected hallucinated and non-hallucinated content stimuli for our EEG experiment. We acknowledge the ongoing discourse around the ethical use of LLMs in scholarly writing—particularly regarding transparency, originality, and accountability. We transparently report the use of LLM assistance and reaffirm that all substantive intellectual contributions (e.g. experimental design, data analysis, interpretation) originated from the authors.

