# OpenReview forum: "How do Human Processes AI-generated Hallucination Contents: a Neuroimaging Study"
_ICLR.cc/2026/Conference — Submitted to ICLR 2026_

### Official Review · Reviewer_tiAT · 2025-10-31

**Soundness:** 2
**Presentation:** 3
**Contribution:** 3
**Rating:** 4
**Confidence:** 2

**Summary:**

This paper investigates the neural mechanisms underlying human processing of hallucinated content about multimodal large language models. The authors conducted EEG experiments with 27 participants and analyzed event-related potentials while participants read image descriptions that either contained or did not contain hallucinated words. Additionally, the paper explores the use of EEG signals to train models for predicting whether a text contains hallucinations.

**Strengths:**

This paper introduces cognitive neuroscience methods into MLLM hallucination research and proposes EEG as an implicit feedback signal for hallucination detection or user trust modeling, which has potential practical value.

**Weaknesses:**

Only 27 participants were included, lacking ecological validity in real-world scenarios. The model analysis is limited, and the EEG prediction experiments do not employ advanced neural network architectures. The analysis of hallucination types is insufficient; although entity, relation, and attribute hallucinations are distinguished, fine-grained comparisons are missing, such as the impact of domain knowledge on hallucinations.

**Questions:**

1.Are the data general or do they include content requiring background knowledge?

2.Do participants with different background knowledge show differences in responses to the same hallucinations? If so, at which stage do these differences appear?

3.Do different types of hallucinations (entity/relation/attribute) show differences in ERP patterns?

4.How consistent are participants’ responses, and does repeated exposure to the same stimuli affect their judgments?

---

> ### Author Response · Authors · 2025-11-17
>
> Thank you very much for your reviews and comments on our paper.
>
> > “Are the data general or do they include content requiring background knowledge?”
>
> As stated in Section 2.2 of the manuscript, we specifically selected the **image-text QA task** from the AMBER benchmark[1] in order to minimize bias arising from participants’ varying disciplinary backgrounds. The stimuli **do not require any prior specialized domain knowledge**: all images and the corresponding descriptive texts are self-contained and easily interpretable by participants from diverse backgrounds. For transparency and reproducibility, **the full set of images and texts is available in our code repository**. We believe this design ensures that the data are sufficiently general and not reliant on participant background knowledge.
>
> > “Do participants with different background knowledge show differences in responses to the same hallucinations? If so, at which stage do these differences appear?”
>
> We did recruit participants from a variety of disciplinary backgrounds (e.g., computer science, mechanical engineering, chemistry, and environmental engineering) as stated in Section 2.1. However, this paper aims to study the general effect of AI hallucinations on humans. Therefore, the task was chosen without the requirement of any specialized prior knowledge (see also the example in Figure 1).
> Our findings show that participants with different backgrounds produced **consistent responses to the same hallucinations**: i.e., comparable accuracy in judgment and broadly similar ERP activation patterns. This outcome **aligns with our design intention of controlling the effects of prior knowledge**. Accordingly, we did not observe a clear stage (e.g., early vs. late ERP component) at which significant differences attributable to background knowledge emerged. We believe this result reinforces the generality of the phenomenon under study.
>
> > “Do different types of hallucinations (entity/relation/attribute) show differences in ERP patterns?”
>
> We did perform a preliminary comparison of the three hallucination categories (entity, relation, attribute). As we mentioned in lines 187–188, accuracy did not differ significantly across the categories. Regarding ERP components, our statistical analyses likewise did not reveal robust, category-specific differences under the current granularity of analysis. Accordingly, we have noted in lines 495–499 that further finer-grained analysis is required, and we have designated this as future work. We believe our current treatment is appropriate for the scope of this study.
>
> > “How consistent are participants’ responses, and does repeated exposure to the same stimuli affect their judgments?”
>
> In lines 183–184 we report the average accuracy across all participants. For full transparency, we will publicly release the dataset (with each participant’s responses to each stimulus) after the review process. In the revised manuscript, we have also **added a standard deviation in Section 3.1 and included each participant’s individual number of correct items in Appendix A.4**.
> With respect to repeated exposure, as clarified in Sections 2.3 and Figures 1, each image-text stimulus was presented **exactly once **to each participant, and participants were allowed to **rest as needed**. Therefore, there was no repeated exposure of the same stimulus to the same participant, and thus, no practice or fatigue effect through stimulus repetition is presented in our data.
>
> Thank you again for recognizing our work. If you have any further questions or concerns, we would be happy to continue the discussion. If you find that this response satisfactorily addresses your concerns, we kindly ask you to consider adjusting your review scores accordingly.
>
> [1] Wang, J., Wang, Y., Xu, G., Zhang, J., Gu, Y., Jia, H., ... & Sang, J. (2023). Amber: An llm-free multi-dimensional benchmark for mllms hallucination evaluation. arXiv preprint arXiv:2311.07397.

---

> ### Author Response · Authors · 2025-11-25
>
> With fewer than ten days remaining in the discussion period, we look forward to addressing any further questions you may have and kindly await your prompt feedback.

---

### Official Review · Reviewer_5ErB · 2025-11-01

**Soundness:** 3
**Presentation:** 3
**Contribution:** 3
**Rating:** 6
**Confidence:** 2

**Summary:**

This paper investigates the neural responses of humans to hallucinated content generated by multi-modal large language models (MLLMs). By recording EEG data from 27 participants, the study examines how human brain activity differs when processing hallucinated versus non-hallucinated content. The results suggest that multiple cognitive processes, including semantic integration, inferential processing, and memory retrieval, are involved in recognizing hallucinations. Additionally, the study demonstrates that EEG signals can predict whether content contains hallucinations, with reliable results when participants correctly identify hallucinations.

**Strengths:**

1) The study provides valuable insights into how the human brain processes hallucinated content, contributing to a deeper understanding of human-AI interaction.

2) The use of EEG data allows for the real-time detection of hallucinations, presenting a novel and non-invasive method to study the cognitive effects of AI-generated content.

3) The research shows significant improvements in the model’s ability to detect hallucinations, providing potential applications for improving AI systems' reliability.

**Weaknesses:**

1) The study is limited by its sample size (27 participants), which may not be representative enough to generalize findings to a broader population.

2) The method relies heavily on participants' ability to accurately recognize hallucinations, which could vary significantly across individuals and affect the reliability of the results.

**Questions:**

See weaknesses.

---

> ### Author Response · Authors · 2025-11-17
>
> Thank you very much for your reviews and comments on our paper.
>
> > “The study is limited by its sample size (27 participants), which may not be representative enough to generalize findings to a broader population.”
>
> We fully acknowledge that a larger sample size can enhance the robustness and generalizability of study findings. However, in research involving user experiments with neural‐signal acquisition (EEG/ERP), recruiting large numbers of participants is often constrained by equipment availability, data-collection time, and signal-quality requirements. For example, **recent studies of multi-subject EEG research employed 21, 20, 12, and 24 subjects** [1,2,3,4].
> Furthermore, our pilot study power analyses show that achieving a **statistical power of 0.8 at an α-level of 0.05 required only 19 participants** (related code has been added to our code repository). To ensure robustness and account for potential data loss or additional variability, we recruited 27 participants—well above this threshold—thereby providing sufficient power for detecting the effects reported.
> Therefore, we believe that we have used a sample size that is comparable to existing research and statistically appropriate. We acknowledge the limitations of sample size and have added relevant discussions in Section 6.
>
> > “The method relies heavily on participants' ability to accurately recognize hallucinations, which could vary significantly across individuals and affect the reliability of the results.”
>
> To minimize the influence of individual differences in prior knowledge or specialized training on the task, we deliberately selected an image-text QA task from the AMBER benchmark [5] that **does not require any domain‐specific background knowledge** (see Section 2.2). Participants are only asked whether the textual description and the image content correspond, and no specialist content knowledge is required. For transparency and reproducibility, the full set of images and texts is available in our code repository.
> Furthermore, as shown in Section 2.1, our participants came from a variety of disciplinary backgrounds (e.g., computer science, mechanical engineering, chemistry, and environmental engineering). Yet, as reported, we observed very **similar performance (judgment accuracy) and highly comparable ERP activation patterns across those backgrounds** — this indicates that the individual differences in background knowledge had minimal influence on the task responses. In the revised manuscript, we have also **added a standard deviation in Section 3.1 and included each participant’s individual number of correct items in Appendix A.4**.
> Hence, while we agree that any behavioral/brain‐signal experiment will have some inter‐individual variability in recognition ability, we believe our task design, participant recruitment strategy, and observed data collectively mitigate the concern that background knowledge variance unduly affected our reliability.
>
> Thank you again for recognizing our work. If you have any further questions or concerns, we would be happy to continue the discussion. If you find that this response satisfactorily addresses your concerns, we kindly ask you to consider adjusting your review scores accordingly.
>
> [1] Mentzelopoulos, G., Chatzipantazis, E., Ramayya, A. G., Hedlund, M. J., Buch, V. P., Daniilidis, K., ... & Vitale, F. (2024). Neural decoding from stereotactic EEG: accounting for electrode variability across subjects. Advances in Neural Information Processing Systems, 37, 108600-108624.
>
> [2] Liu, X. H., Liu, Y. K., Wang, Y., Ren, K., Shi, H., Wang, Z., ... & Zheng, W. L. (2024). EEG2video: Towards decoding dynamic visual perception from EEG signals. Advances in Neural Information Processing Systems, 37, 72245-72273.
>
> [3] Khadir, A., Maghareh, M., Sasani Ghamsari, S., & Beigzadeh, B. (2023). Brain activity characteristics of RGB stimulus: an EEG study. Scientific Reports, 13(1), 18988.
>
> [4] Michalkova, D., Parra-Rodriguez, M., & Moshfeghi, Y. (2022, July). Information need awareness: an EEG study. In Proceedings of the 45th international ACM SIGIR conference on research and development in information retrieval (pp. 610-621).
>
> [5] Wang, J., Wang, Y., Xu, G., Zhang, J., Gu, Y., Jia, H., ... & Sang, J. (2023). Amber: An llm-free multi-dimensional benchmark for mllms hallucination evaluation. arXiv preprint arXiv:2311.07397.

---

### Official Review · Reviewer_hPvT · 2025-11-04

**Soundness:** 1
**Presentation:** 1
**Contribution:** 1
**Rating:** 0
**Confidence:** 5

**Summary:**

The paper “How Do Humans Process AI-Generated Hallucination Contents: A Neuroimaging Study” explores how human brains process hallucinated versus non-hallucinated content produced by multimodal large language models (MLLMs). Using EEG recordings from 27 participants engaged in a multimodal image–text matching task, the authors investigate whether distinct neural patterns emerge when humans recognize or fail to recognize hallucinated content. Each trial involved viewing an image and then a sentence generated by Qwen2.5-VL-3B that either accurately described or hallucinated aspects of the image, followed by a yes/no judgment of correctness. EEG data were analyzed via event-related potentials (ERPs) at different latencies (N100, P200, N400, P600). Results show that hallucination words elicit larger ERP amplitudes—especially in N100, P200, N400, and P600—indicating heightened attention, semantic integration effort, and reanalysis processes. However, when hallucinations are not detected (“HalluWrong” cases), no ERP differences are observed, suggesting an absence of neural anomaly response. Additionally, EEG-based machine learning models (SVM, MLP, attention-based) can predict hallucination presence with high accuracy (AUC ≈ 0.94–0.98 within-subject) but only when participants consciously recognize them. The authors interpret these findings as evidence that hallucinations bypass automatic neural error monitoring when undetected, and that EEG may serve as an implicit signal for identifying deceptive AI content.

**Strengths:**

The study makes a novel interdisciplinary contribution, combining AI hallucination research with human cognitive neuroscience. It bridges the gap between computational detection of hallucinations and human neural processing, offering a fresh empirical perspective on how users engage with AI-generated misinformation.

**Weaknesses:**

Despite its originality, the paper’s empirical rigor and interpretive scope have limitations. First, the sample size (n = 27) and limited number of stimuli per condition constrain statistical power, especially for high-dimensional EEG data analyzed across multiple regions and time windows. The ERP analyses rely on repeated-measures ANOVA across many electrodes and latencies but do not report corrections for multiple comparisons (e.g., Bonferroni or FDR), increasing the risk of false positives. No effect sizes or confidence intervals accompany the p-values, and claims of “significant differences” in Table 1 lack magnitude context. The behavioral results (Table 1) also show higher accuracy for hallucination trials, which is counterintuitive and not discussed—raising potential confounds related to task design or salience bias. The HalluWrong condition, key to RQ2, may suffer from low trial counts, undermining reliability of the “no effect” conclusion. The classification task lacks a clear baseline (e.g., random or non-neural features) and does not report significance testing of AUCs against chance levels. Furthermore, while the authors interpret ERP differences as reflecting “neural deception,” this framing risks overstating causal inference—absence of ERP effects could reflect signal noise or attentional variability, not necessarily cognitive “bypass.” Finally, while Section 6 acknowledges some limitations, it underplays ecological validity concerns: the controlled laboratory paradigm with static images differs from real-world interactions with AI systems.

**Questions:**

Statistical robustness should be strengthened by including effect sizes (η²) and multiple-comparison correction. The paper would benefit from reporting the number of HalluWrong trials per participant and the distribution of EEG epochs per condition. Clarify whether the reported “neural silence” was confirmed through equivalence testing or absence of significant differences.

---

> ### Author Response · Authors · 2025-11-18
>
> Thank you very much for your reviews and comments on our paper.
>
> > Sample size
>
> We fully acknowledge that a larger sample size can enhance the robustness and generalizability of study findings. However, in research involving user experiments with neural‐signal acquisition (EEG/ERP), recruiting large numbers of participants is often constrained by equipment availability, data-collection time, and signal-quality requirements. For example, **recent studies of multi-subject EEG research employed 21, 20, 12, and 24 subjects**. [1,2,3,4]
> Furthermore, power analyses on the pilot study data show that achieving **a statistical power of 0.8 at an α-level of 0.05 required only 19 participants** (related code has been added to our code repository). To ensure robustness and account for potential data loss or additional variability, we recruited 27 participants—well above this threshold—thereby providing sufficient power for detecting the effects reported.
> Therefore, we believe that we have used a sample size that is comparable to existing research and statistically appropriate. We acknowledge the limitations of sample size and have added relevant discussions in Section 6.
>
> > Effect Sizes, Multiple-Comparison Correction
>
> Thank you for raising these important points regarding the statistical rigor of our analyses. We have substantially revised Section 3.2.1 to strengthen the robustness and interpretability of our results.
>
> (1) We have added **effect sizes (partial eta-squared, $\eta\_p^2$) for every statistical test where a p-value is presented**. This ensures that readers can evaluate not only the presence of statistical significance but also the magnitude of the observed effects. Across all tests, we found that $\eta\_p^2$ >0.14, which is conventionally regarded as **a large effect size**. This supports the reliability and practical significance of the neural differences observed in our study.
>
> (2) To mitigate the inflation of Type-I error due to repeated testing across brain regions and time windows, we additionally applied **post-hoc False Discovery Rate (FDR) correction** to all pairwise contrasts. These corrected p-values have been incorporated directly into Table 1 and throughout the revised Section 3.2.1. For full transparency, both the original and FDR-corrected p-values are also reported in Appendix A.3 (Tables 3 and 4).
>
> (3) After applying FDR correction, we observed that a small number of effects in specific brain regions were not statistically significant (e.g., N400 of l-temporal). However, these changes do not alter the main conclusions of the paper, and **the key ERP patterns and behavioral trends described in the original manuscript remain consistent** under the more conservative statistical analysis (e.g., various cognitive processes are engaged at extremely fine temporal scales).
>
> > Behavioral accuracy
>
> Your “The behavioral results (Table 1) also show higher accuracy for hallucination trials” makes us confused.
> We suspect there may be a misunderstanding regarding Table 1 and would like to clarify this.
> **Table 1 reports the significance of ERP differences across time windows and brain regions, and it does not contain behavioral accuracy results.** The notation “Hallu > NoHallu” in Table 1 refers specifically to ERP amplitude differences, indicating stronger neural responses to hallucinated words, rather than higher behavioral accuracy.
> The behavioral accuracy values are presented in Section 3.1. Here, we would like to clarify that the slightly higher accuracy for hallucination cases arises from the simplicity of our image–text verification task. Participants only needed to judge whether the text description matched the visual content, which does not require prior expertise. The average accuracies for the NoHallu and Hallu conditions are 81.17% and 87.41%, respectively. **The accuracy difference between them was not statistically significant.**
> Thus, the behavioral results do not indicate a systematic bias or confound, and they do not affect the validity of our neural findings.
>
> [1] Mentzelopoulos, G., Chatzipantazis, E., Ramayya, A. G., Hedlund, M. J., Buch, V. P., Daniilidis, K., ... & Vitale, F. (2024). Neural decoding from stereotactic EEG: accounting for electrode variability across subjects. Advances in Neural Information Processing Systems, 37, 108600-108624.
>
> [2] Liu, X. H., Liu, Y. K., Wang, Y., Ren, K., Shi, H., Wang, Z., ... & Zheng, W. L. (2024). EEG2video: Towards decoding dynamic visual perception from EEG signals. Advances in Neural Information Processing Systems, 37, 72245-72273.
>
> [3] Khadir, A., Maghareh, M., Sasani Ghamsari, S., & Beigzadeh, B. (2023). Brain activity characteristics of RGB stimulus: an EEG study. Scientific Reports, 13(1), 18988.
>
> [4] Michalkova, D., Parra-Rodriguez, M., & Moshfeghi, Y. (2022, July). Information need awareness: an EEG study. In Proceedings of the 45th international ACM SIGIR conference on research and development in information retrieval (pp. 610-621).

---

> ### Author Response · Authors · 2025-11-18
>
> > HalluWrong condition
>
> The lower number of HalluWrong trials is an inherent property of the task rather than a design flaw.
> Because our task **does not require any domain‐specific background knowledge**, error cases for participants naturally occur infrequently.
> Strictly speaking, no statistical test can prove that "the sample size is definitely sufficient." We conducted a power/sensitivity analysis to demonstrate that, with the current number of trials and participants, we have sufficient power that no strong effect exists for HalluWrong vs. NoHallu, while we acknowledge that subtle differences cannot be fully excluded.
> To further quantify the sensitivity of our HalluWrong analysis, we conducted a post-hoc power/sensitivity analysis on the HalluWrong vs. NoHallu contrast. The within-subject effect size was small ($d\_z$ =0.126). The post-hoc power to detect such a small effect at α=0.05 was only 0.31. In contrast, the same analysis shows that our design would have approximately 80% power to detect medium-to-large effects ($d\_z$ ≥0.372). **This indicates that we are confident that there exists no medium-to-large HalluWrong-specific ERP effects, but it remains unknown to very small effects.** Accordingly, we revised the paper by interpreting the null finding for HalluWrong more accurately, see lines 317-323.
>
> > Baseline for the classification task
>
> In the revised manuscript, we have clarified the selection of the baseline for the classification analysis. Because our evaluation metric is AUC, an untrained or randomly initialized classifier provides a natural and theoretically well-defined baseline: 0.5.
> In addition, we have updated Table 2 to include statistical significance tests comparing our model’s AUC against the random-baseline AUC = 0.5. These tests confirm that the classifier’s performance is significantly above chance, supporting the validity of our decoding results.
>
> > ERP interpretion
>
> We agree that the phrase “bypass the brain’s automatic alarm system” in the Introduction and Conclusion was too strong. Our statistical analyses only show that **the HalluWrong and NoHallu conditions do not exhibit large ERP differences at the group level**. The mechanistic explanation behind this phenomenon remains speculative.
> To address this concern, we have revised the manuscript to **adopt more cautious wording**, changing these statements to formulations such as “may reflect” rather than “necessarily reflects.” We now explicitly emphasize that our findings indicate a lack of strong ERP divergence, rather than evidence for a complete “bypass” mechanism.
> Regarding the reviewer’s concern that the ERP patterns might reflect signal noise, we note that our preprocessing pipeline follows standard and widely accepted EEG procedures. Moreover, ERP waveforms are derived from trial-averaged signals, which substantially reduce non-phase-locked noise by construction. Therefore, the observed ERP components are unlikely to be driven by noise. Instead, they reflect **endogenous neural responses** whose amplitude differences simply do not reach the magnitude of the Hallu vs. NoHallu.
> Finally, we have expanded the discussion in the conclusion to clarify that uncovering the deeper cognitive and neural mechanisms behind the weak HalluWrong effects will require future studies with targeted designs.
>
> Thank you again for recognizing our work. If you have any further questions or concerns, we would be happy to continue the discussion. If you find that this response satisfactorily addresses your concerns, we kindly ask you to consider adjusting your review scores accordingly.

---

> ### Author Response · Authors · 2025-11-25
>
> With fewer than ten days remaining in the discussion period, we look forward to addressing any further questions you may have and kindly await your prompt feedback.

---

### Meta-Review · Area_Chair_Ywh2 · 2025-12-22

**Summary:**

This paper explores the neural mechanisms underlying human processing of AI-generated hallucinatory content through EEG experiments and ERP analysis, and investigates the feasibility of predicting hallucinations via EEG signals. The key concerns raised by the non-malicious reviewers include limitations in sample size generalizability, insufficient ecological validity of the experimental paradigm, relatively limited model analysis approaches, inadequate fine-grained exploration of different hallucination types, and potential impacts of individual differences on results. After comprehensively considering the paper’s contributions, the responses to these concerns, and the remaining unresolved issues, the suggested decision is to reject this submission.

**Reviewer Concerns:**

The author’s rebuttal effectively addressed several key concerns raised by the reviewers. For instance, regarding the sample size, the authors justified its appropriateness by referencing similar existing studies and providing power analysis results, supplemented by relevant discussions in the manuscript; for individual differences, the task design was clarified to not require domain-specific knowledge, and consistent performance and ERP patterns across participants with diverse backgrounds were demonstrated, mitigating concerns about the impact of background variability. Additionally, the authors supplemented effect sizes and FDR correction for statistical analyses, clarified the baseline for the classification task and added significance testing, and revised the wording related to ERP interpretation to be more cautious, addressing corresponding statistical rigor and interpretive overstatement concerns. However, some concerns remain outstanding: the ecological validity gap between the controlled laboratory setting and real-world AI interaction scenarios has not been effectively resolved; the model analysis still lacks advanced neural network architectures, and the fine-grained comparative analysis of different hallucination types (entity, relation, attribute) is insufficient; furthermore, the issue of relatively few hallucination words per participant in the sample has not been adequately addressed.

**Reviewer Scores:**

The reviewers’ initial scores reflect their recognition of the paper’s interdisciplinary innovative value, as well as their attention to existing issues such as sample representativeness and ecological validity, which is consistent with the current completeness and room for improvement of the research. It is worth emphasizing that this study focuses on the neural processing mechanism of humans towards AI-generated hallucinatory content, which is an important theme with both theoretical and practical significance. Its interdisciplinary perspective provides novel ideas for the intersection of AI hallucination research and cognitive neuroscience. We encourage the authors to further improve the sample design, deepen the fine-grained analysis of hallucination types, and enhance the ecological validity of the research in accordance with the reviewers’ comments and the improvement directions in their own rebuttal, and look forward to the revised version better presenting the research value.

---

### Decision · Program_Chairs · 2026-01-26

Reject